# mRNA-programmed translation pauses in the targeting of *E. coli* membrane proteins

Nir Fluman[1]*, Sivan Navon[1], Eitan Bibi[2], Yitzhak Pilpel[1]*

[1]Department of Molecular Genetics, Weizmann Institute of Science, Rehovot, Israel; [2]Department of Biological Chemistry, Weizmann Institute of Science, Rehovot, Israel

**Abstract** In all living organisms, ribosomes translating membrane proteins are targeted to membrane translocons early in translation, by the ubiquitous signal recognition particle (SRP) system. In eukaryotes, the SRP Alu domain arrests translation elongation of membrane proteins until targeting is complete. Curiously, however, the Alu domain is lacking in most eubacteria. In this study, by analyzing genome-wide data on translation rates, we identified a potential compensatory mechanism in *E. coli* that serves to slow down the translation during membrane protein targeting. The underlying mechanism is likely programmed into the coding sequence, where Shine–Dalgarno-like elements trigger elongation pauses at strategic positions during the early stages of translation. We provide experimental evidence that slow translation during targeting and improves membrane protein production fidelity, as it correlates with better folding of overexpressed membrane proteins. Thus, slow elongation is important for membrane protein targeting in *E. coli*, which utilizes mechanisms different from the eukaryotic one to control the translation speed.

*For correspondence: fluman.nir@gmail.com (NF); Pilpel@weizmann.ac.il (YP)

**Reviewing editor**: Nahum Sonenberg, McGill University, Canada

## Introduction

The rate of translation elongation varies considerably between different codons of the same gene, with some codons translated slower than the others (*Ingolia et al., 2009*; *Li et al., 2012*). Various factors can modulate the elongation rate. Some of them are sequence elements within the translated mRNA or protein that directly interact with the translational machinery (*Tuller et al., 2010*; *Li et al., 2012*; *Charneski and Hurst, 2013*; *Pechmann and Frydman, 2013*), such as charged amino acids that interact with the ribosome (*Charneski and Hurst, 2013*) or codons that interact with tRNAs (*Tuller et al., 2010*). Other factors act in *trans* and require the engagement of additional factors that can in turn affect ribosome speed (*Walter and Blobel, 1981*; *Shalgi et al., 2013*). Several studies indicate that the elongation rate can modulate protein folding and function (*Kimchi-Sarfaty et al., 2007*; *Zhang et al., 2009*; *Yona et al., 2013*; *Zhou et al., 2013*), but the underlying mechanisms are not yet well understood.

One of the best understood speed control mechanisms is found in the pathway that targets membrane proteins to the membrane translocon. Membrane proteins are extremely hydrophobic and aggregation-prone and are therefore inserted into the membrane co-translationally (*White and von Heijne, 2008*; *Luirink et al., 2012*). To do so, they must be targeted to the translocon early in translation before large polypeptide portions are synthesized (*Lakkaraju et al., 2008*). Targeting of ribosomes translating membrane proteins is mediated by the ubiquitous signal recognition particle (SRP) system (*Bibi, 2011*; *Akopian et al., 2013*). The core machinery is conserved in all organisms and is comprised of the ribonuceoprotein complex SRP and its receptor. The pathway works such that the SRP binds to the first transmembrane segment (TM) that emerges from the ribosome and directs it together with the translocon-associated SRP receptor to the translocon. In eukaryotes, the SRP arrests

**eLife digest** Proteins are built as long chain-like molecules. First, a length of DNA is copied to make a messenger RNA (or mRNA) molecule, which then binds to a large molecular complex called a ribosome. The ribosome reads and translates the code in the mRNA sequence to build a protein chain, which then folds into a specific three-dimensional shape to allow the protein to perform its function.

Many proteins also need to be targeted to the right location within the cell in order to carry out their role. Some proteins have to be inserted into the membranes of cells and these proteins are directed, as they are being built, to the membrane by another molecular complex called the signal recognition particle (or SRP for short). The SRP binds to the new protein as it emerges from the ribosome and helps to direct it to the membrane. To make sure that membrane proteins fold correctly, their translation is paused whilst the protein is being targeted to the membrane. Plants, animals, and other eukaryotes do this via a unique part of the SRP complex that only allows the translation to continue once the translating ribosome has been brought close to the membrane. However, most bacteria lack this part of the SRP complex, and yet they are still able to accurately insert new, correctly folded, proteins into their membranes. This suggests that an alternative mechanism must exist in bacterial cells.

Fluman et al. looked at an existing data set that had measured how many ribosomes are found at different points along the length of mRNA molecules at any given time in the bacterium *E. coli*. If ribosomes are consistently found at specific sites in given mRNA molecules, it suggests that these are the sites where a pause in translation occurs. Specific short mRNA sequences—that bind to a ribosome and hold it in place—are often found in these pause sites. These sequences are similar to another sequence, called the Shine–Dalgarno sequence that is often also found at the very start of an mRNA molecule, where it functions to recruit a ribosome and begin the translation process.

Fluman et al. reveal that mRNAs of membrane proteins contained these similar sequences early on in their coding region. Some looked likely to pause the translation before the newly formed protein chain emerged from the ribosome, which could give the ribosome time to be targeted to the membrane. Other Shine–Dalgarno-like sequences were found slightly later on in the mRNA molecules for protein chains that span back-and-forth through the membrane several times.

Fluman et al. show that slowing translation in this manner—which is different to that used by eukaryotes—helps to ensure that membrane proteins are folded correctly in *E. coli*. Although these pauses occur frequently, mainly in the early stages of the translation of membrane proteins, there are many other translation pause sites that are known to exist in other mRNAs. The next challenge is to understand the function of these other pause sites, and how they work together with other mechanisms to regulate translating ribosomes inside cells.

the elongation upon binding to ribosome-nascent chain complexes, in order to prevent the protein synthesis before targeting is complete (*Walter and Blobel, 1981*; *Mason et al., 2000*). The arrest is mediated through the SRP Alu domain that binds to the elongation factor binding site (*Halic et al., 2004*) and is kept until the ribosome arrives at the translocon, causing the SRP dissociation. Puzzlingly, however, the bacterial world displays a dichotomy in the structure of the SRP, wherein Gram-positive bacteria typically possess an Alu domain and Gram-negatives, such as *E. coli*, are typically devoid of it (*Regalia et al., 2002*). Accordingly, in vitro translation studies suggested that the *Escherichia coli* SRP cannot arrest translation (*Raine et al., 2003*). Thus, other mechanisms that compensate for the lack of Alu-mediated arrest in Gram-negative bacteria could be required, especially considering their faster translation rates compared to eukaryotes (*Young and Bremer, 1976*; *Bonven and Gulløv, 1979*). One such putative mechanism has recently been described (*Yosef et al., 2010*), which calls for further investigation.

Recently, our ability to measure translation rates was revolutionized by the technology of ribosome profiling, enabling the estimation of elongation rates on a genome-wide level and at a single codon resolution (*Ingolia et al., 2009*). Studies conducted so far revealed remarkable heterogeneity in ribosome density between different codons within genes and suggested that translational pausing or slowing are actually quite common events (*Ingolia et al., 2011*; *Li et al., 2012*). These experiments also greatly advanced our understanding of how sequence elements modulate elongation rate (*Li et al., 2012*; *Charneski and Hurst, 2013*). A notable example is found in *E. coli*, where many if not most of

the pause sites are mediated by internal Shine–Dalgarno-like elements, which are capable of transiently binding to the ribosome as it translates the mRNA (*Li et al., 2012*). Despite this progress, it is still not fully understood what could be the function of such commonly observed pauses in cell physiology.

To meet this challenge, we explored where Shine–Dalgarno-mediated pause sites are preferentially located across the *E. coli* transcriptome, by analyzing the existing data from ribosome profiling (*Li et al., 2012*). Our analyses revealed significant enrichment of pause events during the initial stages of membrane proteins translation. This location coincides with the stage in translation in which membrane proteins are thought to be targeted to the membrane translocons. We provide evidence that Shine–Dalgarno-mediated pauses in *E. coli* possibly compensate for the lack of SRP Alu domain-mediated arrest during membrane protein targeting.

## Results

### Programmed pauses are enriched at the beginning of *E. coli* membrane proteins

Recently, the ribosome occupancy across *E. coli* mRNAs was profiled in a genome-wide manner, revealing the locations of ribosome slowing as peaks of high ribosomal density. In that study, *Li et al. (2012)* found that ribosome pause sites are often explained by a Shine–Dalgarno (SD)-like sequence located 8–11 nucleotides upstream of the currently decoded codon. In order to understand the potential function of pauses better, we sought to investigate where such mRNA-programmed pauses are preferentially located. To identify pause sites, we searched for codons exhibiting high *local* ribosome density compared to other codons in the same coding sequence. Ribosome density profiles for well-expressed genes (i.e., with reliable ribosome profiling data, 'Materials and methods') were calculated by normalizing the ribosome density in codons to the median density of the gene (*Figure 1—figure supplement 1A*). Unless otherwise noted, all analyses were based on these normalized profiles. Assuming that ribosomes flow along mRNAs is in steady-state, ribosome density in these profiles is inversely proportional to translation speed (*Tuller et al., 2010*). The first and last four codons were excluded since they contain high density related to an initiation 'ramp' and translation termination, respectively (*Tuller et al., 2010*; *Oh et al., 2011*). We considered 'pause codons' as those that constitute the 5% of the most dense codons genome-wide, corresponding to ~sixfold higher density than the median density of genes in which they reside (*Figure 1—figure supplement 1A*). To generate a dependable SD-mediated set of pause sites, we filtered out pause codons for which we could not identify an upstream SD-like element (*Figure 1—figure supplement 1B*, see 'Materials and methods'). Codons that passed both filters (high local density and SD-like element) constitute ~2.5% of all codons and we refer to them as 'programmed pauses'.

We first compared the pause occurrence of three protein classes: cytosolic, (inner) membrane, and periplasmic proteins. The ribosome density profiles of all coding sequences were aligned at their start codon, and the fraction of programmed pause codons in each position was analyzed. The results show that membrane proteins are enriched with pauses between codons 16 and 60, after which they level-down to values similar to cytosolic proteins (*Figure 1A*). Pauses occur in roughly 4% of the proteins at each codon position in membrane proteins within the codon range 16–60 (*Figure 1A*). Indeed, over the entire range of codons from position 16 to 60, we find that 69% of membrane proteins have at least one programmed pause position, significantly higher compared to cytosolic (50%, $p = 1.5 \times 10^{-7}$) and periplasmic proteins (41%, $p = 6 \times 10^{-8}$, Z-test for 2 proportions) (*Figure 1A*, inset). The occurrence of pauses in this codon range indicates that pauses are specifically enriched in membrane proteins only in the early stages of translation elongation, in a coding sequence region which coincides with the stage of membrane protein targeting. Two distinct peaks of elevated pauses were observed, the first in codons 16–36 and the second in codons 40–60 (defined here as 'region I' and 'region II', respectively, *Figure 1A*). In the following sections, we study each of these peaks separately.

An important concern was that the pauses may occur as by-products of amino-acid or codon composition biases near the N-terminus of membrane proteins. In *Figure 1—figure supplement 2*, we exclude several such biases for membrane proteins, by showing that positively-charged residues, glycine codons, or rare codons are not significantly enriched in codons 16–60 in membrane proteins. Notably, however, such abundant biases in periplasmic proteins may explain the apparent depletion of pause in their N-terminal codons (*Figure 1A*, *Figure 1—figure supplement 2*), likely due to the presence of N-terminal signal peptides.

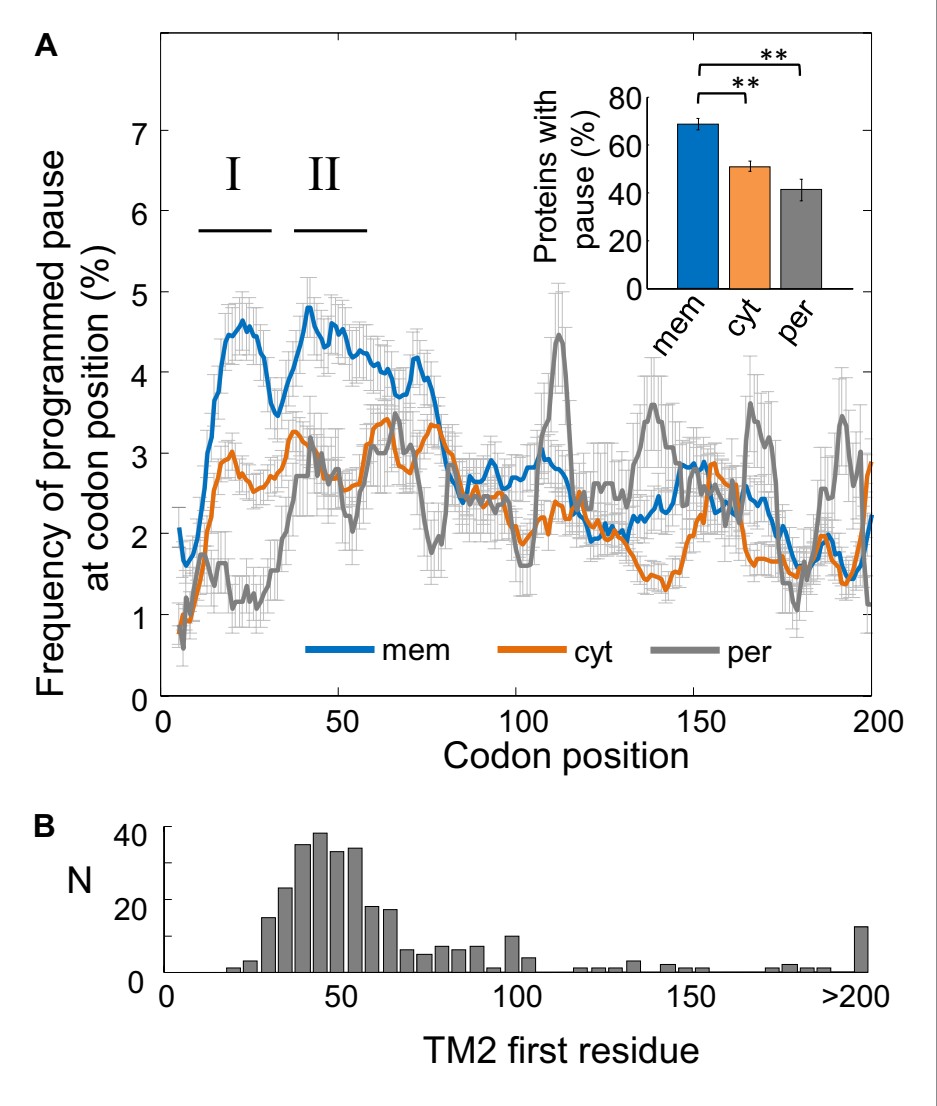

**Figure 1**. Frequency of programmed pauses across coding sequences. (**A**) Percentage of the programmed pauses in every codon across the coding sequences in membrane, cytoplasmic, and periplasmic proteins. Regions of elevated pause in membrane proteins are marked by I and II. Pause codon positions refer to codons in the A-site of the ribosome. Inset: percentage of proteins from different classes having at least one pause in the codon range 16–60. ** indicates $p < 10^{-6}$. Error bars in figure and inset indicate s.e. for proportion. (**B**) Histogram of the distribution of the positions of the first codon of TM2 in *E. coli* membrane proteins that were analyzed in (**A**). N, number of proteins in which TM2 starts at the indicated position.

The following figure supplements are available for figure 1:

**Figure supplement 1**. Detection of mRNA-programmed pauses.

**Figure supplement 2**. Amino acid and codon biases are not mediating pauses in membrane proteins' codons 16–60.

## Pausing of ribosomes before the translation of the second transmembrane segment

We first investigated the programmed pause enrichment in region II, around codons 40–60 (*Figure 1A*); region I is discussed in a later section. We noticed that region II coincides with the location of the beginning of the second transmembrane segment (TM2) in many membrane proteins (*Figure 1B*). To test a potential link, we aligned all membrane proteins according to their TM2 start (*Figure 2A*) and

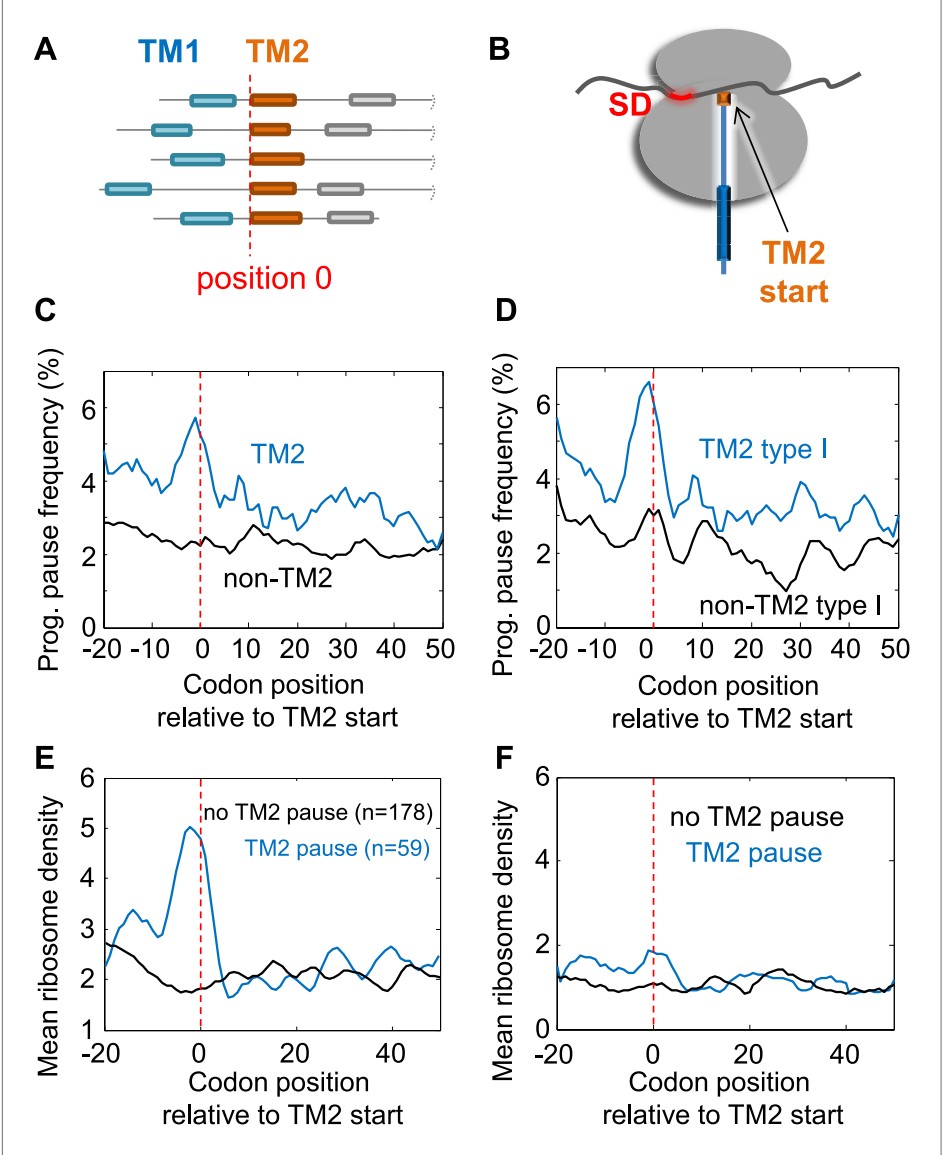

**Figure 2**. Pause before translation of TM2. (**A**) Membrane proteins were aligned such that the first residue of TM2 is aligned to position 0 (red line). (**B**) Visual scheme of the stage at which pause occurs during translation. Red marks SD-like element in the mRNA. Blue line and cylinders depict nascent polypeptide and possible TM location in the tunnel, respectively. (**C** and **D**) The frequency of codons having programmed pauses in every position was analyzed, either in all membrane proteins (**C**) or only in proteins having intracellular N-termini (**D**). Profiles of proteins aligned to TM2 are in blue. Profiles of proteins aligned to all other TMs are in black. (**E**) Average ribosome density profiles of proteins with (blue) or without (black) identifiable programmed pauses in codons −5 to +1 relative to TM2 start. (**F**) Same as (**E**) but only in mRNA nucleotide positions lacking upstream SD-like elements.

The following figure supplements are available for figure 2:

**Figure supplement 1**. Specificity of programmed pause before TM2.

**Figure supplement 2**. Comparison between proteins with and without identifiable pauses before TM2.

calculated the frequency of pauses along the aligned coding sequences at various distances from that point. Indeed, we found a peak of high pause frequency just before the start of TM2 (*Figure 2C*) ($p = 3 \times 10^{-6}$, 2-sample *t* test), extending from five codons before TM2 to the second codon of the TM.

This observation suggests that membrane protein translation is programmed to pause just before TM2 residues are added to the elongating nascent chain (*Figure 2B*). Curiously, the peak is much sharper when considering only proteins with intracellular N-termini (*Figure 2D*) (p = 4 × 10$^{-9}$, 2-sample *t* test), in which TM2 occurs in 'Type-I topology' (the N-terminus of TM2 is extracellular), suggesting that proteins with extracellular N-termini do not pause before TM2. To explore the difference between the two groups, we analyzed amino acid biases before TM2 (*Figure 2—figure supplement 1A*). The main difference was excess of arginines and lysines before the TM in proteins with extracellular N-termini, in accordance with the 'positive inside rule' (*von Heijne, 2006*). This difference is predicted to increase pauses in this location (*Charneski and Hurst, 2013*), unlike the actually observed decreased pauses in this group of proteins. Thus, differences in amino-acid sequences cannot account for the difference in the pausing behavior. We attempted to explore if proteins with extracellular N-termini may instead have pauses at other locations, however, analyses of these proteins were precluded by low statistical power due to small sample size (only 88 proteins in our data set are predicted to have extracellular N-termini, as opposed to 237 with intracellular N-termini). Therefore, with the current limitations, we cannot conclude if these proteins may have pauses elsewhere or if they do not utilize the pause strategy altogether. We therefore excluded these proteins, which constitute a minority of the membrane proteome, from further analysis in sections that deal with pauses in region II.

Notably, when considering only the remaining proteins set (i.e., with intracellular N-termini), the peak of pause before TM2 could not be explained by amino-acid biases at this location since the slightly enriched amino acids there (*Figure 2—figure supplement 1A*), are not encoded by G-nucleotide-rich codons that resemble the SD motif. Furthermore, the peak of pause in the proteins was specifically related to TM2, as we could not find similar peaks when aligning the coding sequences according to other TMs (*Figure 2—figure supplement 1B*).

The pauses occur 35 ± 13 codons after the beginning of the first TM. Given that the exit tunnel of the ribosome admits ~28 amino acids of the extended polypeptide (*Bornemann et al., 2008*), pausing of the ribosome at these positions means that in most proteins the first TM has partially emerged out of the ribosome during pausing. The pause location is reminiscent of the location at which the translation arrest occurs in the eukaryotic case, when the SRP arrests translation after TM1 was synthesized and exposed out of the ribosome. This suggests a common physiological role of slowing elongation during targeting. Interestingly, only ~25% of membrane proteins (59 out of 237) have identifiable pauses at codons −5 to +1 relative to TM2 start, and indeed only these proteins display high local ribosome density in this region (*Figure 2E*). The quantitative difference in density between proteins with and without pause in *Figure 2E* suggests that the pause duration is approximately 1.5 s (roughly the time it takes to translate 28 codons, see 'Materials and methods'). To check the dependence of translation rate on SD-like elements, we repeated the density analysis of *Figure 2E* with a modified data set in which all nucleotide positions with even weak upstream SD-like motif were excluded from the analysis ('Materials and methods'). The results show that upon SD exclusion, the slow-down in this region was nearly abolished (*Figure 2F*) suggesting that the SD motif is necessary to generate the observed pauses. It is important to note that several important pauses probably escaped our detection. First, the pauses may occur at subtly different locations from codon −5 to +1 relative to TM2 and our thresholds for pause consideration might be too stringent. Secondly, the TMs and their boundaries are not accurately predicted by the existing methods (in about 25% of proteins, we found disagreements of at least 10 residues in the location of TM2 between two databases, Uniprot and ExTopoDB). Nevertheless, it seems that SD-programmed pauses before TM2 occur only in a subset of the membrane proteome and are not crucial for the production of all membrane proteins.

We thus hypothesized that proteins with or without a pause before TM2 may have different characteristics. Our analyses revealed two interesting differences. First, proteins with pause had higher expression on an average (p = 0.004, two sample *t* test) (*Figure 2—figure supplement 2A*). This is consistent with a role of the pauses in promoting the production fidelity, since highly expressed proteins and their mRNAs require more tight quality control and are hence better adapted for fidelity (*Drummond and Wilke, 2008*; *Tuller et al., 2010*; *Levy et al., 2012*). Secondly, we find that TM1 in proteins with pause is less hydrophobic compared to proteins without identifiable pauses, suggesting that proteins with pause may have compromised affinity for SRP (*Lee and Bernstein, 2001*), (p = 0.006, two sample *t* test) (*Figure 2—figure supplement 2B*). This is consistent with a recent study showing that slow elongation rates may rescue defects in the SRP-mediated targeting of membrane proteins with sub-optimal targeting signals, likely by extending the time window for productive targeting (*Zhang and Shan, 2012*).

## Long loops between TM1 and TM2 are an alternative solution to pause before TM2

The existence of pauses before TM2 in only a subset of membrane proteins suggests the existence of alternative solutions in other proteins. We hypothesized that the role of the pause before TM2 is to delay the synthesis of TM2 after TM1 was synthesized, in order to complete membrane-targeting before synthesis of additional hydrophobic protein segments. If this is correct, then a long loop between these TMs whose translation requires more time could be just as beneficial. Indeed, our analysis revealed that first loops (i.e., those between TM1 and TM2) are enriched for long lengths of ≥60 residues and depleted of short lengths of ≤10 residues (*Figure 3A*, *Figure 3—figure supplement 1A*) (p = 6 × 10⁻¹⁰ and p = 3 × 10⁻³ respectively, hypergeometric test). Consecutive loops locations between the other TMs showed no significant enrichment for longer loop (*Figure 3—figure supplement 1B*). Although there could be alternative explanations for why membrane proteins evolved to have larger first loops, such loops can certainly delay the translation of TM2 after TM1 was synthesized. Indeed, we found that proteins having long first loops (≥60 residues, 34 proteins) almost never had pauses before TM2 compared to proteins with shorter loops (*Figure 3B*) (p = 0.01, hypergeometric test). These findings suggest that the coding sequences of membrane proteins are designed to delay the translation of TM2 after TM1 was synthesized, either by extending the loops connecting these TMs or by pausing.

## Pausing of ribosomes before the nascent peptide emerges out of the tunnel

We next investigated the pause in codons 16–36 (region I, *Figure 1A*). The frequency of programmed pause codons and the mean codon density in this range were computed separately for the membrane and cytosolic proteins. Programmed pauses were enriched in membrane proteins compared to cytosolic proteins (p = 4 × 10⁻⁸, Z-test for two proportions) (*Figure 4A*). Consequently, local ribosome densities in membrane proteins mRNAs were significantly higher than those of cytosolic proteins (p = 1 × 10⁻³⁶, 2-sample *t* test) (*Figure 4C*), suggesting slower translation. A mean difference of 0.56 density units per codon (*Figure 4C*) over a range of 21 codons indicates that translation is slowed by

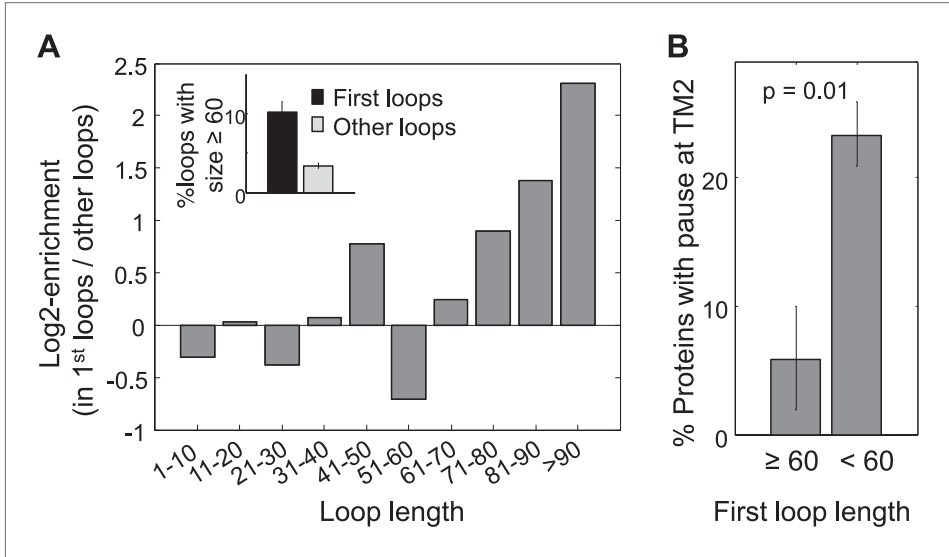

**Figure 3**. Enrichment of long first loops and its effect on pausing. (**A**) Enrichment of various loop lengths in first loops compared to all other loops. Shown are Log2 values of ratios. Inset: occurrence of long loops of length ≥60 in first loops or other loops. Error bars indicate s.e. for proportion. (**B**) Occurrence of pause before TM2 in proteins with long or short first loops. Error bars indicate s.e. for proportion.
The following figure supplement is available for figure 3:

**Figure supplement 1**. Analysis of loop lengths.

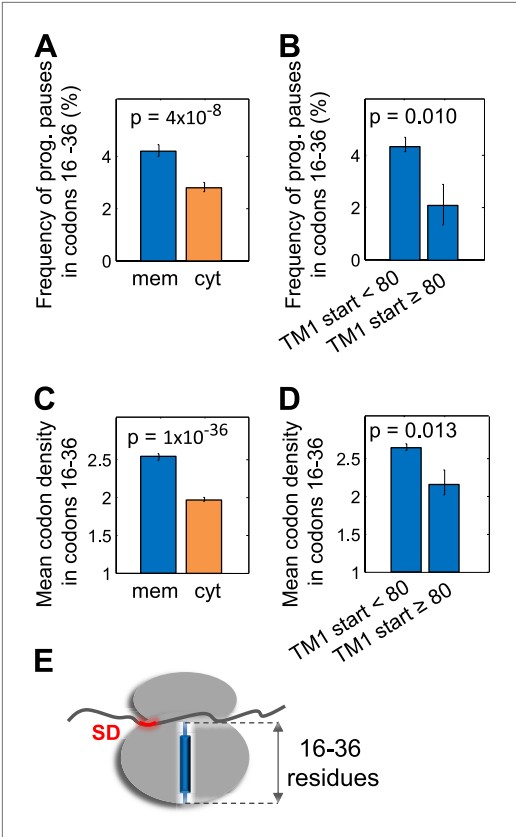

**Figure 4**. Pause in codon range 16–36. (**A** and **C**) Comparison of the frequency of programmed pauses (**A**) or mean ribosome density (**C**) in codon range 16 to 36 between cytoplasmic and membrane proteins. Error bars indicate s.e. for proportion (**A** and **B**) or SEM (**C** and **D**). (**B** and **D**) Same comparisons between membrane proteins in which the first residue of TM1 occurs after or before position 80 of the polypeptide. (**E**) Visual scheme of the stage at which pause occurs during translation, similar to **Figure 2B**.

~0.6 s (the time required to translate 12 codons, see 'Materials and methods') compared to the average cytosolic protein.

A tendency to pause between codons 16 and 36 indicates that membrane proteins are programmed to pause before the nascent polypeptide emerges out of the ribosome exit tunnel (**Figure 4E**). In this state, the SRP cannot directly interact with the nascent peptide. However, recent studies showed that *E. coli*'s SRP can efficiently target such ribosome-nascent-chain complexes to the membrane (**Bornemann et al., 2008**; **Holtkamp et al., 2012**) and that this is done regardless of the class of translated proteins (cytosolic/membrane/periplasmic). Thus, increased pausing at this codon range may perhaps give membrane proteins an advantage over cytosolic proteins in being targeted even before direct SRP–nascent peptide interaction is possible.

To further test this hypothetical link to the SRP pathway, we considered also a later stage of the pathway in which the peptide emerges from the exit tunnel. A previous study showed that once a sufficiently long peptide emerges, targeting by the *E. coli* SRP becomes substrate-specific; namely, if the exposed peptide contains a hydrophobic transmembrane segment, then SRP will remain bound to the ribosome-nascent-chain complex and direct its targeting. Otherwise, hydrophilic peptides cause SRP-dissociation (**Bornemann et al., 2008**). We therefore reasoned that membrane proteins having long hydrophilic N-terminal tails before TM1 may constitute a unique case and may not benefit from such pauses, since the emerging hydrophilic peptide would cause SRP-dissociation and productive targeting would occur only later, when the first TM emerges. Indeed, unlike other membrane proteins, proteins with at least 80 residues before TM1 (n = 16) did not show an elevated pause frequency at codons 16–36 (p = 0.01, Z-test for two proportions), nor did they show elevated ribosome density (p = 0.013, one-tailed two-sample *t* test) (**Figure 4B,D**). Thus, although further studies are necessary, our results suggest that pauses that occur when the peptide is still unexposed may aid in the targeting of membrane proteins without long hydrophilic N-terminal tails.

## Slowly translated proteins are better folded when overexpressed from plasmid

Our findings suggest that slow translation might be beneficial during the targeting of *E. coli* membrane proteins, but what is the function of these pauses? A role in promoting production fidelity is suggested by the function of translation arrest in eukaryotes. In order to test this hypothesis directly, we chose a set of 100 membrane proteins to overexpress in *E coli*. Based on the results above, proteins having long polypeptide stretches (>50 residues) before the first TM or between TM1 and TM2 were excluded from the expression set. Overexpressed proteins were hybrids with a C-terminal GFP-His$_8$, which enabled the separate detection of both the unfolded and folded protein levels, as shown for several membrane proteins (**Waldo et al., 1999**; **Geertsma et al., 2008**; **Linares et al., 2010**; **Müller-Lucks et al., 2012**; **Schlegel et al., 2012**). The discrimination is possible because unfolding or aggregation of fused proteins hinders GFP-folding and -fluorescence, whereas folded and fluorescent

GFP is indicative of the functional and folded fused protein (*Geertsma et al., 2008*; *Linares et al., 2010*; *Müller-Lucks et al., 2012*; *Schlegel et al., 2012*). The resistance of folded GFP to SDS-denaturation allows the resolution of both GFP forms by SDS-PAGE and Western blotting, since folded GFP migrates faster (*Figure 5—figure supplement 1A*). To confirm the method, we assessed the detergent-solubility of five different overexpressed membrane proteins. The results of fractionation studies showed that only the folded GFP-coupled forms were soluble with mild detergent, while the unfolded forms were insoluble and probably aggregated (*Figure 5—figure supplement 1A*).

Next, we expressed the entire set of proteins and quantified both the folded and unfolded forms (see an example in *Figure 5—figure supplement 1B*). Eleven proteins were excluded from the original set due to problematic detection and two additional ones were outliers (*Figure 5—source data 1*). Our set of 87 tested proteins covers a wide range of translation speeds at regions I and II, that is, at codons 16–36 and before TM2 (*Figure 5A*). We divided the set to three quantiles according to translation speed: from the 33% of genes having the fastest translation speed (lowest ribosome density) to those having the slowest speed (high density) in these codon ranges (*Figure 5A*, colors). Remarkably, genes with slower translation at these codons showed a 95% elevation in expression of the folded protein form, compared with genes having fast translation (p = 3 × 10⁻⁴, two-sample *t* test) (*Figure 5B*). This difference was not observed for expression of the unfolded/aggregated form of the proteins (*Figure 5C*), suggesting that translation speed during targeting specifically influences the extent of correct folding rather than affecting the overall expression level.

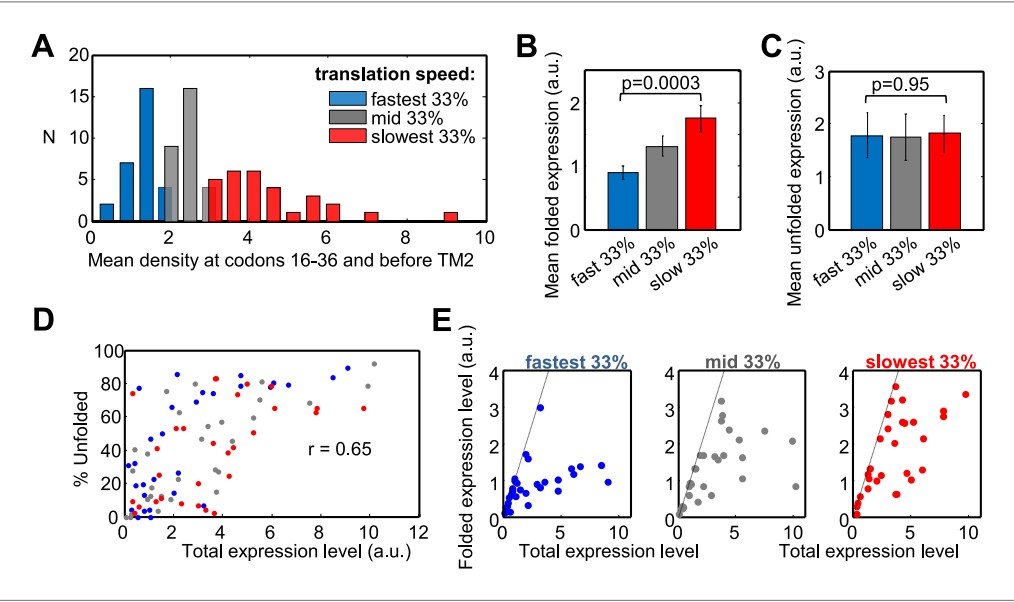

**Figure 5**. Effect of translation rate at codons 16–36 and before TM2 on membrane protein overexpression. (**A**) Histogram of translation speeds. For each gene, the ribosome density at codons 16–36 and before TM2 was averaged. The distribution of values in the different genes is shown (N, number of genes with indicated density). Colors indicate groups corresponding to different translation speeds at these codon ranges and accompany the rest of the figure. (**B** and **C**) Mean (±SEM) expression values of the folded (**B**) and unfolded (**C**) forms of proteins from the three different groups. (**D**) Effect of the total expression level on the percentage of unfolding in the different protein groups. (**E**) Folded vs total levels of expression in the three protein groups. Each dot represents a different protein. Dashed line depicts a 1:1 ratio wherein all of the expressed protein is detected in its folded form. The experiment was repeated three times.

The following source data and figure supplements are available for figure 5:

**Source data 1**. Quantification of protein overexpression.

**Figure supplement 1**. Detection of folded and unfolded membrane protein–GFP-His₈ hybrids.

**Figure supplement 2**. Effect of translation rates at various positions on protein overexpression.

We then turned to investigate how expression level affects folding and its potential interplay with the speed of translation in determining the quality of folding. The percentage of unfolded protein was significantly correlated with the total level of the protein (folded + unfolded) (Spearman's r = 0.65, p = 6 × 10$^{-12}$, *Figure 5D*). This indicates that regardless of the identity of the overexpressed protein, the cell's capacity to produce high amounts of folded membrane proteins is limited, and increased expression levels would result in increased aggregation. We propose that this dependency reflects saturation of the membrane protein biogenesis machineries in the cell, which becomes less efficient upon high membrane protein overexpression. This finding is in agreement with previous indications that membrane proteins are often better expressed in functional form under lower induction regimes (*Wagner et al., 2008*).

Consistent with this notion, we observe that as total protein levels increase, the levels of folded proteins approach saturation (*Figure 5E*). However, the saturation curves show remarkable differences between proteins that are translated with different speed regimes in our codon range of interest. Proteins that are translated fast saturated at relatively low amounts of folded protein (apart from a single protein), while slowly translated proteins converged to higher saturation levels, if they saturated at all (*Figure 5E*). This effect is contributed by translation slowing at both codons 16 to 36 and at codons before TM2 (*Figure 5—figure supplement 2A*). The trend is also reflected by the difference between the groups in *Figure 5D* and suggests that slow translation during targeting supports less aggregation of the overexpressed proteins. Notably, slow translation at other more downstream codon locations may also be beneficial for folding (*Figure 5—figure supplement 2B*). However, slow translation is much more common at codons 16–60, that is at the codon region that covers regions I and II, than downstream (*Figure 5—figure supplement 2B*), suggesting that slowing during targeting is of particular importance.

Overall, the results suggest that regardless of the protein identity, slow translation during targeting is associated with better folding. Yet, the effect appears to depend on expression level and is possibly required only when membrane protein biogenesis becomes saturated. We propose that the presence of pauses in many proteins may alleviate the burden on cellular quality control under endogenous expression levels.

## Engineered SD-like motifs improve the folding of membrane proteins

In order to test the potential causative relationship between slow translation and better folding, we set out to modify the rate of translation of genes by modifying their SD-like elements. In our limited set of genes, we found that engineering SD-like motifs in genes that are translated fast would be easier than tampering with existing SDs in slow genes. Three genes that are translated fast in our codon range of interest were selected, and 12 or 14 mutations were designed for each gene in order to maximize the affinity of the mRNA for the ribosomal anti-SD, while maintaining the amino-acid sequence unaltered (*Figure 6—figure supplement 1*). The silent mutations enabled robust increases of anti-SD binding affinity in only two genes, ygdD and brnQ, while for the third gene, ybjJ, it only permitted a moderate increase (*Figure 6A*). In order to test the effect of the mutations on protein folding, we implemented the GFP-folding assay. Indeed, upon overexpressing the wild-type and mutant genes, we observed that the two mutants with robust increase in SD motifs folded better (*Figure 6B,C*, YgdD and BrnQ), whereas the third gene showed no difference (*Figure 6B,C*, YbjJ). Thus, SD-like motifs improve membrane protein folding and prevent aggregation, likely by slowing-down translation.

## Translation pauses in targeting of *Bacillus subtilis* membrane proteins

Our findings indicate that, like in eukaryotes, slow translation during targeting promotes the fidelity of membrane protein production and folding, yet the underlying mechanisms are different. It seems that *E. coli* compensates for the lack of SRP Alu domain-mediated arrest, at least in part, by strategically positioned SD-mediated pauses. To test the notion of compensation more directly, we turned to analyze the data on *B. subtilis*, an organism that can potentially utilize both mechanisms. SD-like elements were shown to control translation pauses in *B. subtilis*, like in *E. coli* (*Li et al., 2012*). Additionally, although SRP-mediated arrest was never demonstrated in *B. subtilis*, unlike *E. coli*, this organism may have all components required for SRP-mediated arrest: the Alu domain of SRP RNA is present and this domain binds a protein (HBsu) that may function analogously to the eukaryotic SRP14 (*Nakamura et al., 1999*).

In order to examine the pausing behavior of *B. sublitis*, we analyzed ribosome profiling data of this organism (*Li et al., 2012*). We deliberately analyzed pause events, irrespective of the presence of

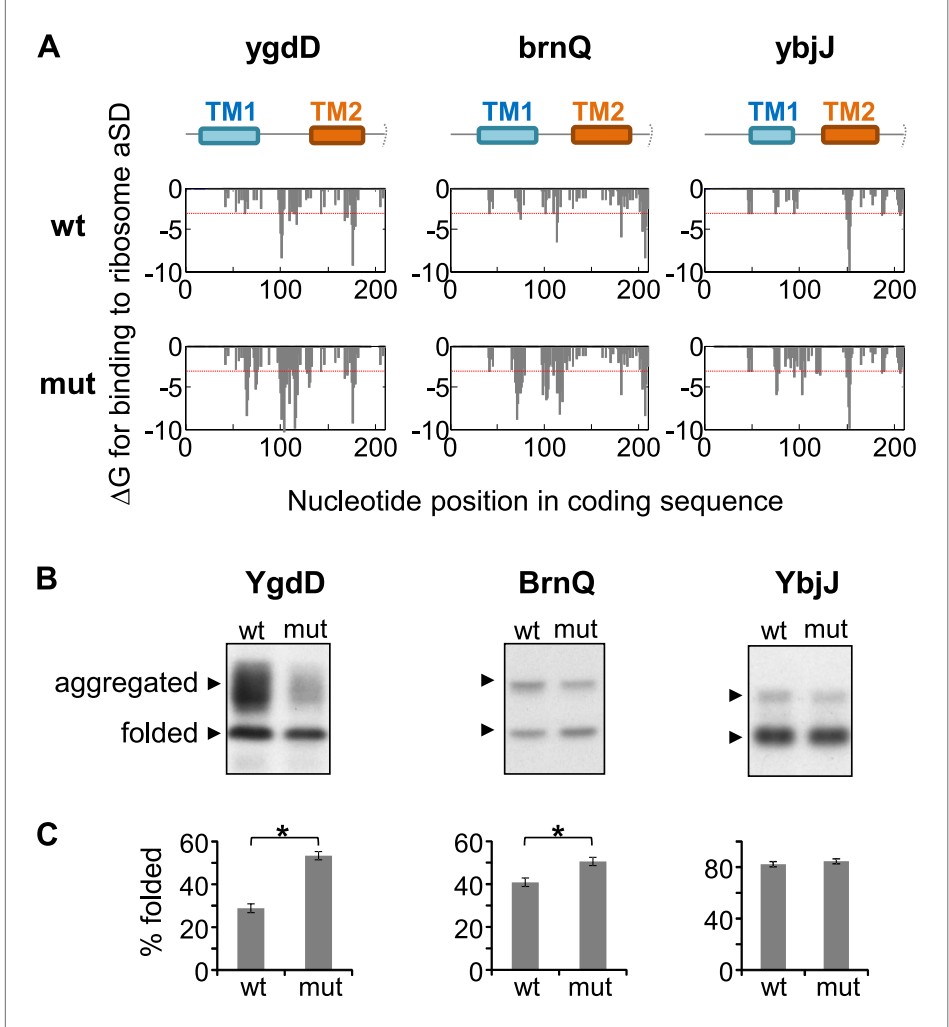

**Figure 6**. Effect of silent mutations introducing SD-like motifs on protein folding and aggregation. (**A**) Effect of mutations on affinity for ribosomal anti-Shine–Dalgarno sequence. Lower panels show calculated binding energies to anti-Shine–Dalgarno sequence, similar to *Figure 1—figure supplement 1B*. wt, wild-type; mut, mutant. Upper panel indicates the nucleotide positions in the coding sequence that code for TM1 and TM2. Note that all proteins and coding sequences are longer than the range of 70 amino-acids and 210 nucleotides shown. (**B**) Western blot analysis of protein–GFP-His8 hybrids, similar to *Figure 5—figure supplement 1*. (**C**) Densitomentry-based quantification of the percentage of the folded state from the total protein. Error bars represent SEM from three independent experiments. * indicates p < 0.01 (one tailed, paired *t* test).

The following figure supplement is available for figure 6:

**Figure supplement 1**. Silent mutations introducing SD-like motifs.

SD-like sequences, since we suspected that in this bacterium pauses might not necessarily be programmed into the mRNA sequence. Thus, all codons having high local densities (within the top 5%) were counted as pauses. First, we compared membrane and cytosolic proteins by aligning their coding sequences to their start codons and analyzing the fraction of pause codons in each position. *Figure 7A,B* show that in both *B. subtilis* and *E. coli*, pauses were enriched in membrane proteins compared to cytosolic proteins; however, the precise positioning at which the pauses occurred was different. While in *E. coli* the enrichment occurred around codons 15–65, in *B. subtilis* it occurred only downstream around codons 45–85, with a peak at codon 64. Notably, the positioning of transmembrane segments along the sequence was comparable between *E. coli* and *B. subtilis* (*Figure 7A,B*, orange bars and *Figure 7—figure supplement 1*).

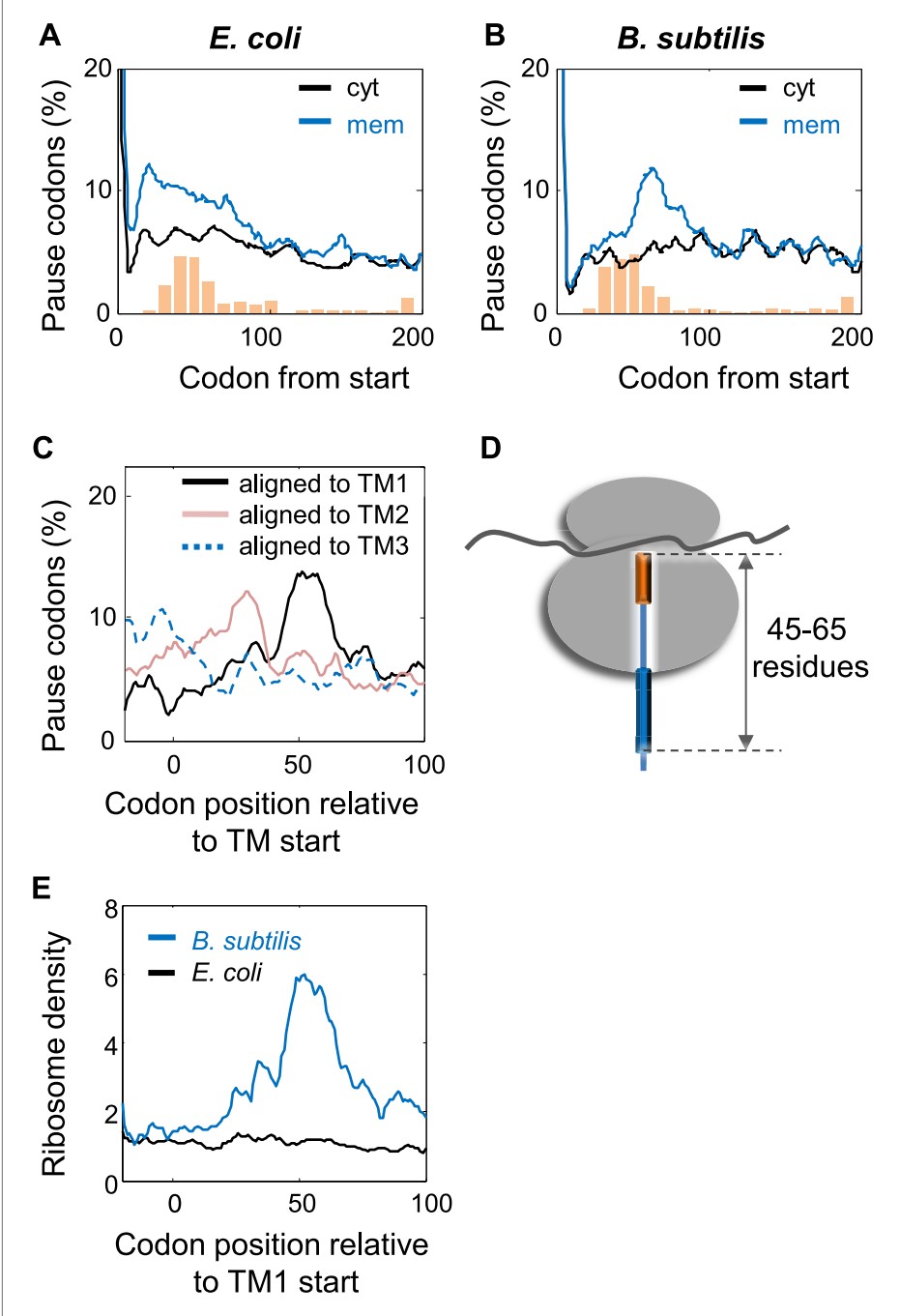

**Figure 7**. Comparison of pausing events in *B. subtilis* and *E. coli*. (**A** and **B**) Percentage of pauses in codons across coding sequences in membrane (mem) and cytoplasmic (cyt) proteins in *E. coli* (**A**) and *B. subtilis* (**B**). Orange histograms at the bottom indicate the distribution of the position of the first codon of TM2 (similar to *Figure 1B*). (**C**) *B. subtilis* membrane proteins were aligned such that the first residues of TM1 or TM2 or TM3 are aligned to the position 0. The percentage of codons having pauses in every position was analyzed. (**D**) Visual scheme of the stage at which pause occurs during translation in *B. subtilis*, similar to *Figure 2B*. (**E**) Effect of position relative to TM1 on ribosome densities in positions lacking upstream SD-like elements.

The following figure supplement is available for figure 7:

**Figure supplement 1**. Histograms of TM locations in *B. subtilis* and *E. coli*.

The different positioning indicates that unlike *E. coli*, the pauses in *B. subtilis* were not localized to the regions at or before TM2 (*Figure 7B*). We suspected that the Alu domain-mediated arrest is functional in *B. subtilis*, and therefore the pauses are expected to occur upon direct interaction of SRP with the first TM, when the TM emerges out of the ribosome exit tunnel. To examine this, we aligned the coding sequences of *B. subtilis* membrane proteins to the positions of the first residues of TM1 and the average pause profile was analyzed. This analysis showed a high elevation of pause events that occurred roughly in 45–65 codons after synthesis of the first residue of TM1 (*Figure 7C,D*). Notable, albeit lower, pause peaks were also observed when aligning the coding sequences to TM2 and TM3 (*Figure 7C*). However, considering that the distance between TM1 and TM2 or TM3 is not very variable (median distances 35 and 72, respectively), these peaks most likely reflect shifted and skewed representations of the peak downstream to TM1.

The position of the pause 45–65 codons downstream to TM1 agrees well with the exposure of the first TM out of the ribosome, where it can directly interact with the SRP (*Houben et al., 2005*). If this pause is indeed mediated by the Alu domain, then it should operate regardless of SD-like elements. To test SD-independent pauses, we excluded all positions in the sequence that have even weak upstream SD-like elements ('Materials and methods'). Using this SD-depleted data set, we then checked how the translation rate (as judged by the local ribosome density) is affected by the position relative to TM1. The results show that ribosome density 45–65 codons downstream to TM1 was significantly elevated compared to other positions (*Figure 7E*), suggesting that *B. subtilis* utilizes a pause strategy that is independent of the SD-like sequences, possibly involving the SRP Alu domain. This trend was not observed in *E. coli* (*Figure 7E*), suggesting that in the absence of SD-like sequences, the ability of the *E. coli* SRP to arrest elongation is low, if any. Taken together, our results suggest that *B. subtilis* employs the eukaryotic-like strategy for slowing the elongation of translated membrane proteins, and consequently does not utilize SD-mediated pauses to this end.

## Discussion

Membrane proteins constitute roughly 25% of the proteomes in all living organisms (*Wallin and von Heijne, 1998*) and their hydrophobic nature necessitates co-translational insertion. The ubiquitous SRP system is responsible for the timely delivery of ribosomes translating membrane proteins to membrane translocons. The conservation of the Alu domain in both eukaryotes and Gram-positive bacteria suggests that slow elongation during targeting is universally important (*Mason et al., 2000*; *Lakkaraju et al., 2008*). Therefore, the lack of this domain in Gram-negative bacteria, such as *E. coli*, was puzzling since it was not clear if and how such organisms can slow the elongation during targeting. In principle, the control of ribosome speed could be encoded in *cis* in the translated mRNA, or in *trans*, for example as in the case of the SRP Alu domain. Recent in vivo studies indicated that the *E. coli* SRP can inhibit membrane protein translation, but the precise underlying mechanism and the potential for influencing elongation rate are not yet clear (*Yosef et al., 2010*). Our analyses of in vivo translation rates in *E. coli* and *B. subtilis* support the notion that strong SRP-mediated arrest that is SD-independent requires the Alu domain and is therefore probably missing in Gram-negative bacteria. Remarkably, our results suggest that *E. coli* depends, at least in part, on a *cis*-encoded means to attenuate translation during targeting, presumably due to the lack of the Alu domain. Other bacteria probably also need to compensate for the lack of Alu domain-mediated translation arrest, and the utilization of SD-mediated pauses might be a general solution.

The mRNA-programmed mechanism is reminiscent of the Alu domain-mediated one. Both mechanisms act to slow elongation in the early stages of translation, in order to facilitate membrane protein-folding and avoid aggregation. However, there are also notable differences. First, the position of the pauses in *E. coli* is different from the position of Alu domain-mediated arrest, with the former occurring earlier in translation. The reason for this is currently unclear, but it might be related to the potential for earlier targeting of ribosomes in *E. coli*, even before the nascent peptide emerges from the ribosome exit tunnel (*Bornemann et al., 2008*; *Holtkamp et al., 2012*). Secondly, while the prokaryotic pausing relies on information embedded in the mRNA, the eukaryotic counterpart reads the protein. Lastly, there is a principal strategic difference between the two mechanisms. The SRP-mediated arrest is only relieved after proper targeting and SRP-dissociation, thus ensuring the fidelity of the process (*Akopian et al., 2013*). In contrast, at present it is reasonable to guess that the mRNA-programmed mechanism probably pauses for a fixed amount of time, without any feedback about the targeting status. In that respect, the SRP-mediated mechanism appears to be more efficient and

elegant. Notably, however, it remains to be studied whether SD-mediated pauses may also be influenced by external factors that may report on the translocon arrival, as shown for translation arrest-peptides (*Ismail et al., 2012*). An attractive possibility is that SD-mediated pauses may somehow cooperate with the bacterial SRP and utilize the feedback provided by SRP-dissociation upon completion of targeting. Such a mechanism may also explain how the *E. coli* SRP regulates translation (*Yosef et al., 2010*).

Slow translation is not essential for membrane–protein targeting. In eukaryotes, the arrest function can be abolished by mutation without severely compromising viability (*Mason et al., 2000*; *Lakkaraju et al., 2008*). Similarly, in *E. coli* we observe a notable variability in the rate of translation between different membrane proteins. Our results indicate that slow translation is important especially at elevated protein amounts, when the cell's membrane protein biogenesis capacity might approach saturation. This observation agrees well with recent results in mammalian cells showing that slow translation helps in overcoming rate limitations that arise due to low concentrations of the SRP receptor. It therefore seems that both eukaryotes and prokaryotes possess limiting amounts of membrane protein biogenesis machineries, which may frequently be saturated, creating rate-limiting bottlenecks. These saturation problems are alleviated by slowing down the translation in strategic locations, which allows more time for targeting to complete before further polypeptide synthesis. In a broader perspective, translation pauses are a general strategy that is employed when a process that requires only a portion of the polypeptide is hindered by additional polypeptide segments. Two such processes are the co-translational targeting and the folding of individual domains in the multidomain proteins (*Zhang et al., 2009*; *Pechmann et al., 2013*). The time separation provided by the pause allows the completion of these processes without interruption, thus helping to avoid problems in protein folding and aggregation.

Finally, we note that our findings may have implications to membrane protein research in general. Heterologous expression of eukaryotic membrane proteins in *E. coli* is notoriously difficult, often leading to poor expression of functional proteins, creating a serious bottleneck in the field (*Granseth et al., 2009*). Our study adds to recent studies revealing the existence of sequence preferences within coding sequences of membrane proteins in *E. coli* (*Prilusky and Bibi, 2009*; *Nørholm et al., 2012*, *2013*), which may need to be obeyed by heterologous coding sequences. For example, SD-mediated pauses are probably lacking a priori in heterologous coding sequences. It therefore seems that in that respect, Gram-positive bacteria such as *B. subtilis* might be more compatible with eukaryotic membrane protein heterologous expression (*Kunji et al., 2005*), since they contain a translation arrest logic similar to the eukaryotic one.

## Materials and methods

### Data

Genome sequences and associated data were downloaded from the NCBI FTP site (accessions NC_000913 for *E. coli* and NC_000964 for *B. subtilis*). Ribosome profiling data were taken from NCBI GEO accession GSE35641 (samples GSM872393 and GSM872397).

### Protein localization and prediction of transmembrane segments

Predicted locations of TMs for *E. coli* were from ExTopoDB (*Tsaousis et al., 2010*). For *B. subtilis*, TMs were predicted using the TMHMM algorithm (*Krogh et al., 2001*). Only proteins having at least two TMs were considered as membrane proteins. Cytosolic and periplasmic protein localizations were taken from Uniprot.

### Identification of pause events

Ribosome density data were obtained from *Li et al. (2012)*.Only genes with sufficiently high ribosome density reads were analyzed (median density of reads across the coding sequence >0.2). For each gene, the ribosome density in every nucleotide was divided by the median ribosome density across the coding sequence. Codon-wise density was calculated by averaging the nucleotide density in every three nucleotides of each codon. These densities indicate ribosome occupancy when the respective codons are located in the A-site of the ribosome. The first and last four codons were excluded from further analysis since they contain high densities associated with the slow translation initiation and termination. The threshold for pause codon consideration (top 5% of codon-wise density) was calculated by collecting codon density from the entire genome (for *E. coli* and *B. subtilis*, the thresholds were 6.06 and 23.4-fold higher than the median, respectively). Programmed pauses in *E. coli* were

defined as codons having density above these thresholds and also having at least one SD-like element 8–11 nucleotides upstream of at least one of the codon nucleotides. Shine–Dalgarno-like elements were identified based on the calculated binding energy of 10 nucleotide-long sequences to the ribosomal anti-SD, as described (*Li et al., 2012*). All positions having calculated binding energy of ΔG < −3.1 kcal/mol, which constitute the lowest 5% (genome-wide), were considered as SD-like elements. We note that these thresholds represent a compromise since even higher ΔG values may affect the ribosome density, and in addition many places with lower binding energies show no indication of pause (see *Figure 1—figure supplement 1C*).

## Estimation of translation time

To estimate differences in translation time, we calculated how much extra ribosome density is associated with a particular set of genes compared to a control set of genes over a specific range of codons. For codon range 16–36, we compared membrane and cytosolic genes over codons 16–36. A mean difference of 0.56 density units per codon (*Figure 4C*) over a range of 21 codons yields a difference of 11.8 density units. For codons before TM2, we computed the difference between proteins having or lacking programmed pauses at codons −5 to +1 relative to TM2 start. Mathematically, the difference is equal to the area trapped between the two curves in *Figure 2D* and equals 28 density units. Since the density units in the genes' profiles are normalized to the median codon density of the gene, it is assumed that each single density unit corresponds to the median time it takes the ribosome to translate a single codon. To estimate translation time, we used a previously determined value of elongation rate for the *E. coli* ribosome of 1 codon per 0.058 s (*Young and Bremer, 1976*).

## Generation of SD-depleted data sets

All mRNA positions having calculated the binding energy of ΔG < 2 kcal/mol were considered potential SD-like elements. Accordingly, ribosome densities 8–11 nucleotides downstream of them were excluded from analysis.

## Expression levels

The median ribosome density of each gene from *Li et al. (2012)* was taken as expression level. Compared with mRNA levels, this quantity correlates better with protein expression as it reflects both RNA levels and the level of translation from each transcript (*Ingolia et al., 2009*).

## TM hydrophobicity

Sequences corresponding to the first TM were analyzed. TMs shorter than 24 residues were increased to 24 by symmetrically including flanking residues from both sides. A sliding window average of hydrophobicity was calculated with a window length of 18 residues using the Kyte–Doolittle hydrophobicity scale (*Kyte and Doolittle, 1982*). Only the highest hydrophobicity score was kept for each TM.

## Heterologous expression

Plasmids encoding membrane proteins–GFP-His$_8$ hybrids were previously reported (*Daley et al., 2005*). Only proteins having cytosolic N- and C-termini (*Daley et al., 2005*) were analyzed. The plasmids were transformed to *E. coli* BL21(DE3)pLysS. Overnight cultures grown in LB with kanamycin (30 µg/ml) and chloramphenicol (34 µg/ml) were diluted 1:100 into fresh (same) medium and allowed to grow for 2.75 hr at 37°C, until they reached a typical OD$_{600}$ of 0.4–0.6. Cultures were then induced with 0.4 mM IPTG and allowed to grow for additional 2 hr and transferred to ice. Cells were harvested by centrifugation at 4°C, washed once with ice-cold buffer A (50 mM Tris–HCl pH 8, 100 mM NaCl, 1 mM EDTA), resuspended in the same buffer at an OD$_{600}$/ml of 1.5–2, and kept at −20°C. For disruption, cells were thawed, supplemented with 1 mM phenylmethanesulfonyl fluoride (PMSF), and sonicated well until the solution cleared completely.

## Fractionation of membranes and aggregates

The procedure was carried at 4°C, and the samples were kept from all fractions. The crude sonicated cells (0.8 ml) were ultracentrifuged at 200,000×*g* for 30 min. The pellet was thoroughly resuspended in 0.72 ml of buffer A supplemented with 1 mM PMSF and then solubilized by the addition of 80 µl 10% β-dodecylmaltoside (Anatrace) and homogenized extensively. Samples were incubated for 30 min on ice and then re-centrifuged at 200,000×*g* for 30 min. Portions corresponding to equal amount of

the starting material were taken from each fraction, supplemented with 1:4 of 5× SDS sample buffer (120 mM Tris–HCl pH 6.8, 50% glycerol, 100 mM DTT, 2% SDS, and 0.1% bromophenol blue), and loaded to 10% SDS-PAGE for analysis.

## Western blotting and in-gel fluorescence
In-gel fluorescence was recorded using Molecular Dynamics Typhoon 9400 (GE Amersham). The gels were subsequently transferred to nitrocellulose membranes for Western analysis. His-tagged proteins were detected with His probe-HRP (Pierce).

## High throughput quantification of membrane protein aggregation
Cultures were grown in 96-well plates with each clone in a single well. Each growth experiment included two independently grown cultures harboring the emrD gene as internal standards. Sonicated cells (0.2 ml) overexpressing membrane proteins were mixed with 0.05 ml 5× SDS sample buffer and loaded to 10% SDS-PAGE gels. Each gel included the two emrD standard clones. The gels' proteins were detected for in-gel fluorescence and Western blotting (see above). Densitometric quantifications were carried using ImageQuant TL software (GE Healthcare, Tel-Aviv, Israel). First, the amount of the folded protein was extracted from fluorescence measurements and normalized to that of the folded EmrD (the amount of folded EmrD protein was defined as 1 arbitrary unit). The fraction of folded and aggregated protein for each clone was calculated based on quantification of Western blots. The amounts of aggregated and total protein were calculated by combining the knowledge on amount of the folded protein (from fluorescence) with the fraction of the folded protein (from Western blotting). Quantification values are reported in the *Figure 5—source data 1*.

## Mutagenesis
Mutagenesis was achieved by PCR with 5′-phosphorylated mutagenic primers (Sigma-Aldrich) that amplify the entire plasmid. The PCR products were blunt-end ligated and transformed to *E. coli* DH5α. Mutant plasmids were sequenced to verify that they contain only the desired mutations. The primers sequences were: ygdD mutagenesis (GGGCGGTGGAGATGGGGTGGATACAGACGGGGCTCGAAT ACCAGGCGTTTC and CCATCGTCTTACTCAACACATGCGCCCCAAACGCCCCCAGAGCCACAAAA ATGAAG); brnQ mutagenesis (CCTCCAATGGTGGGGTTGCAGGCGGGGGAGCACGTGTGGACT GCGGCATTCG and GAAAATAATGTTCCCCGCCCCCACAAACAACGCAAATGTCATAAAGCC); ybjJ mutagenesis (GGGCGACGAGGACGCCGGCGATAAGGGATATTCTCTCTGTCTCG and ACGACGCC ATCAACAACCCCGGCAAAAAGAAGAACATAA).

## Acknowlegements

We thank Gunnar von Heijne and Daniel O Daley for generously providing the plasmid library. We thank the Pilpel lab and Maya Schuldiner for useful discussions. NF acknowledges the Clore Foundation for scholarship support. YP acknowledges the European Research Council for grant support (grant tRNAProlif, # 616622). YP holds the Ben May Professorial Chair.

---

## Additional information

### Funding

| Funder | Grant reference number | Author |
| --- | --- | --- |
| European Research Council | 616622 | Nir Fluman, Sivan Navon, Yitzhak Pilpel |
| Clore Duffield Foundation | postdoctoral fellowship | Nir Fluman |

The funders had no role in study design, data collection and interpretation, or the decision to submit the work for publication.

---

### Author contributions
NF, Conceived the study, Designed all experiments and analyses, Acquisition of data, Analysis and interpretation of data, Drafting or revising the article; SN, EB, Analysis and interpretation of data; YP, Designed all experiments and analyses, Analysis and interpretation of data, Drafting or revising the article

## Additional files

### Major datasets

The following previously published datasets were used:

| Author(s) | Year | Dataset title | Dataset ID and/or URL | Database, license, and accessibility information |
|-----------|------|---------------|------------------------|--------------------------------------------------|
| University of Wisconsin | 2013 | E. coli genome | http://www.ncbi.nlm.nih.gov/assembly/GCF_000005845.2/ | Publicly available at NCBI. |
| BSNR | 2009 | B. subtilis genome | http://www.ncbi.nlm.nih.gov/assembly/?term=NC_000964.3 | Publicly available at NCBI. |
| Li G, Oh E, Weissman JS | 2012 | The anti-Shine-Dalgarno sequence drives translational pausing and codon choice in bacteria (ribosome profiling data) | http://www.ncbi.nlm.nih.gov/geo/query/acc.cgi?acc=GSE35641 | Publicly available at NCBI Gene Expression Omnibus (Samples GSM872393 and GSM872397). |

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
