## [Decision Letter]

Thank you for sending your work entitled “mRNA-programmed translation pauses in the targeting of *E. coli* membrane proteins” for consideration at *eLife.* Your article has been favorably evaluated by Aviv Regev (Senior editor) and 3 reviewers, one of whom is a member of our Board of Reviewing Editors.

Fluman and colleagues describe a model how translation of membrane proteins is regulated in *E. coli* to avoid misfolding and aggregation of these proteins. This is an interesting and long-standing question, since the *E. coli* SRP lacks the Alu domain and it is thus unclear whether SRP can influence the elongation rates of membrane proteins. In a recent study, Yosef and co-authors (MBio, 2010) showed that overexpression of the M-domain of Ffh selectively affected translation of membrane proteins suggesting that Ffh might be responsible for arresting elongation in bacteria. In the present study Fluman and co-authors propose another mechanism of translational pausing based on the analysis of ribosome profiling data obtained by Li et al. (Nature, 2012). The authors suggest that in *E. coli* elongation pausing during membrane protein synthesis is programmed in the mRNA and mediated by the presence of *Shine-Dalgarno*-like (SD-like) sequences upstream of the TM2 coding region. Pausing at this position would allow concomitant binding of *E. coli* SRP to TM1. This mechanism applies to 25% of the membrane protein encoding mRNAs whereas synthesis of the remaining membrane proteins would be regulated by other mechanisms such as long internal loop sequences between TM1 and TM2.

All the reviewers agree that the paper is interesting, that the computational study is very thorough, and that the results are well presented and suggest an original model. As such it could stimulate research in this area. They also commented that the main conclusion is well reasoned and makes good physical sense. Being mainly computational, the manuscript states the caveats clearly and does not try to oversell the observations.

Nonetheless, the findings are largely correlative in nature. The main concern about the observed correlation between internal *Shine-Dalgarno* sites and transmembrane domains is that, because of the nature of the genetic code, both features have higher propensity towards a subset of amino acids, and non-causal correlation may arise from such effects. For example, glycine-rich regions tend to have more *Shine-Dalgarno*-like sites (G-rich nucleotide sequences) and to be found in between transmembrane domains (flexible loops). Additionally, among the synonymous codons for the same amino acid, *Shine-Dalgarno*-like codons tend to be rare codons. Together with the propensity to have rare codons towards the N-terminus of proteins, this might explain the increased number of pauses at the 5' end and the fact that there is diminished pausing at the second transmembrane domain when the second transmembrane domain is farther from the N-terminus (and therefore longer loops in between 1st and 2nd TM). To make a compelling case that evolution selects specifically for SD-like sites to facilitate membrane targeting, the authors need to establish that the enhanced pausing is not simply a by-product of the biases in amino acid or codon choices in this region.

The paper would be substantially strengthened if a causal link were established between the presence of SD-like sequences and improved folding and expression of membrane proteins. A direct experiment is to surgically remove the pauses and monitor the change in the folding efficiency and membrane targeting. This could be accomplished by introducing synonymous mutations to remove SD-like sequences and use the authors' reporter system to see if there is an enhanced defect in folding or membrane localization. It would nice if in the future you could remove the Alu portion of *B. subtilis* SRP RNA in the gram positive bacteria *B. subtilis*, to see if there is corresponding change in translational pausing and folding. For this paper however, you need to emphasize that the idea vis-a-vis *B. Subtilis* is highly hypothetical.

In summary, the results presented in this manuscript have potentially important implications for both the coding of bacterial proteins and the translocation mechanism for membrane proteins. A few key additional analysis and experiments as outlined above would remove ambiguity in the interpretation of the results and substantially strengthen the conclusions.

Some specific comments:

1) A concern is the lack of quantitative data on multiple occasions. For instance, it is mentioned that “To generate a dependable, SD-mediated set of pause sites, we filtered out pause codons for which we could not identify an upstream SD-like element”, however it is not clear what was the number (or percentage of total) of the filtered-out codons. Similarly, in the Materials and methods section the authors note that “these thresholds represent a compromise since even higher ΔG values may affect ribosome density and in addition, many places with lower binding energies show no indication of pause”, but unfortunately no data are presented as to what is the number or ratio as whole of these non-fitting occurrences. There are a few more cases of this sort throughout the manuscript. The authors need to clarify these aspects of the report and possibly discuss the statistical significance of the elimination of the non-fitting occurrences and its impact on the final conclusion.

2) While the authors excluded the proteins with TM2 that have extracellular N-termini (despite forming a sizable fraction of the total membrane proteins, 25%), it is not discussed why these proteins don't resemble their intracellular N-termini counterparts in the translational pause mechanism. If the pause is required for prevention of aggregation of the intracellular N-termini proteins, why is it not necessary for the extracellular ones?

3) The following is confusing. The authors first assign the peak of pause to the TM2 of the intracellular N-termini proteins. Yet, when discussing the possible irrelevance of the positive-inside rule in this case it is stated, “the N-termini of these TMs are actually extra-cellular”. One can't figure out whether the subject is still the intracellular N-termini protein or extracellular ones. On a related note, it would be interesting to investigate whether or not any particular type of amino is enriched in the N-termini of the extracellular ones.

4) Where it is stated that “only ∼22% of membrane proteins (69 out of 321) have identifiable pauses at codons -5 to +1 relative to TM2 start”. This doesn't seem to be consistent with the earlier statement that 69% of the membrane proteins have at least 1 pause site between codon 16-60. Although it is not clear how many of these membrane proteins have either TM1 or 2 or both, one could infer from Figure 1 that the frequency of pause at TM1 and 2 is more or less 50:50. What could explain the discrepancy between 22% and 34.5% (half of 69%)?

5) As a negative control in Figure 2, the ribosome densities aligned to TM2 for membrane protein-encoding mRNAs without SD-like sequences should be shown (the ones that were filtered out). Like in Figure 6 for the TM1 aligned sequences.

---

## [Author Response]

We are pleased to submit a revised version of our manuscript for consideration at *eLife*. We addressed all major concerns as outlined below. In particular we did the requested experiment, in which we seeded *Shine-Dalgarno* sites in three genes, right in the place where it often naturally appears. Reassuringly, two of the genes showed the expected increase in folding efficiently, thus establishing a causal link beyond the originally observed correlation.

*Nonetheless, the findings are largely correlative in nature. The main concern about the observed correlation between internal* Shine-Dalgarno *sites and transmembrane domains is that, because of the nature of the genetic code, both features have higher propensity towards a subset of amino acids, and non-causal correlation may arise from such effects. For example, glycine-rich regions tend to have more* Shine-Dalgarno*-like sites (G-rich nucleotide sequences) and to be found in between transmembrane domains (flexible loops). Additionally, among the synonymous codons for the same amino acid, Shine-Dalgarno-like codons tend to be rare codons. Together with the propensity to have rare codons towards the N-terminus of proteins, this might explain the increased number of pauses at the 5' end and the fact that there is diminished pausing at the second transmembrane domain when the second transmembrane domain is farther from the N-terminus (and therefore longer loops in between 1st and 2nd TM). To make a compelling case that evolution selects specifically for SD-like sites to facilitate membrane targeting, the authors need to establish that the enhanced pausing is not simply a by-product of the biases in amino acid or codon choices in this region.*

We agree that this is an important point. We now analyzed amino acid biases and the N-terminal bias for rare codons. We show that the bias for rare codons is strong only before codon 16 and is already diminished in codon range 16-60 where the pauses in membrane proteins occur. Additionally, the rare codons bias shows no difference between membrane and cytosolic proteins (Figure 1—figure supplement 2), indicating that it cannot explain the difference between these protein groups. Our results indicate that there is no specific bias for Gly residues in the region where we see abundant pauses in membrane proteins (codons 16-60, Figure 1—figure supplement 2) and here again, there is virtually no difference between membrane and cytosolic proteins. Amino-acid biases in regions preceding TM2 are also now analyzed in Figure 2—figure supplement 1 and we observe no bias towards Gly or any other amino acid that is encoded by G-rich nucleotide sequences. Moreover, we now show that in 60% of the cases, SD-like elements that mediate the pauses are placed such that the sequence does not necessarily encode Gly residues because it is present also out of the reading frame, supporting direct selection for the nucleotide motif rather than for particular amino acids (Figure 1—figure supplement 2). Together, the results alleviate the concern that SD-mediated pauses in membrane proteins are by-products of biases in codon or amino-acid choices.

Importantly, however, during these analyses we learned that there are abundant biases in the N-terminal portion of periplasmic proteins (Figure 1—figure supplement 2), where we claimed that a depletion of pauses exists in the original manuscript. These biases are most likely due to the presence of signal peptides in this location. Specifically, the depletion of pauses could arise as a by-product of depletion of Gly residues. Therefore, it is unclear if the pause depletion in periplasmic proteins is of functional significance and we have thus deleted the parts in Figure 4 that originally dealt with this group of proteins as well as the 5 lines in the original text describing these parts. We note that this periplasmic proteins portion is only peripheral to the manuscript, and that we did not pursue it further, and therefore it should not undermine the main message of the paper, which deals with membrane proteins.

*The paper would be substantially strengthened if a causal link were established between the presence of SD-like sequences and improved folding and expression of membrane proteins. A direct experiment is to surgically remove the pauses and monitor the change in the folding efficiency and membrane targeting. This could be accomplished by introducing synonymous mutations to remove SD-like sequences and use the authors' reporter system to see if there is an enhanced defect in folding or membrane localization*.

We have now done this experiment, with a minor modification. Instead of tampering with existing SD sites from genes that contain pauses, we built new SD sites in genes that lack pauses and suffer from a folding defect (new Figure 6). The reason for building and not tampering with SD sites is technical: with the few target genes to choose from (whose expression is sufficiently high), we found that we were not able to make a large impact on the SD-sites of genes that contain pauses using only synonymous mutations. On the other hand, in two genes that lack pauses we could greatly enhance several SD motifs without modifying the amino-acid sequence. Indeed, these genes showed enhanced folding compared to wild-type, suggesting a causative relationship between pausing and proper folding. In a third gene, where we could only introduce weak SD motifs we did not see such an effect. The results are reported in Figure 6 and its supplement and are described in a new section entitled “Engineered SD-like motifs improve the folding of membrane proteins”.

*It would nice if in the future you could remove the Alu portion of* B. subtilis *SRP RNA in the gram positive bacteria* B. subtilis*, to see if there is corresponding change in translational pausing and folding. For this paper however, you need to emphasize that the idea vis-a-vis* B. Subtilis *is highly hypothetical.*

We replaced the statement “strongly suggesting that the Alu domain-mediated pause is operational in *B. subtilis* and is independent of SD-like sequences” from the original manuscript with “suggesting that *B. subtilis* utilizes a pause strategy that is independent of SD-like sequences, possibly involving the SRP Alu domain.”

*1) A concern is the lack of quantitative data on multiple occasions. For instance, it is mentioned that “To generate a dependable, SD-mediated set of pause sites, we filtered out pause codons for which we could not identify an upstream SD-like element”, however it is not clear what was the number (or percentage of total) of the filtered-out codons. Similarly, in the Material and Methods section the authors note that “these thresholds represent a compromise since even higher ΔG values may affect ribosome density and in addition, many places with lower binding energies show no indication of pause”, but unfortunately no data are presented as to what is the number or ratio as whole of these non-fitting occurrences*.

We have addressed this by showing three examples for different ΔG thresholds in the new Figure 1—figure supplement 1 and by modifying the Results and Methods sections.

*There are a few more cases of this sort throughout the manuscript*.

The data are in Figure 1. We now modified the figure legend to make this clearer. We modified the legends of several other figures to make their quantitative nature more clear.

The relevant data now states: “only ∼25% of membrane proteins (59 out of 237) have identifiable pauses at codons -5 to +1 relative to TM2 start.”

*The authors need to clarify these aspects of the report and possibly discuss the statistical significance of the elimination of the non-fitting occurrences and its impact on the final conclusion*.

The best way to address this is to simply analyze translation speed, as calculated from raw ribosome profiling data, without considering any thresholds. This was addressed in Figures 2 and 4 in the original manuscript (in the new submission, Figures 2 and 4). We now also added Figure 2 that addresses this point.

Nevertheless, to alleviate any concern, we repeated the major analyses of Figures 1 and 2 with another energetic threshold for SD filtering, where every sequence with a predicted binding energy ΔG < -2.2 for the ribosomal aSD would be called an SD-like motif. This threshold is set to capture 10% of the coding sequence as potential SD-like motifs, unlike the previous threshold of 5% (ΔG < -3.1; see the new Figure 1—figure supplement 1). We add the new figures to the response letter and not to the main manuscript as we do not feel that they contribute much to the paper (Figure 8).Author response image 1.

The figures show the same trends as the original ones, with the exception that in the modified Figure 1 the double-peak behavior appears somewhat weaker. Nevertheless, it is exactly this double-peak behavior that led us to discover the peak of pause before TM2, which is also confirmed here with the new threshold in the modified Figure 2.

*2) While the authors excluded the proteins with TM2 that have extracellular N-termini (despite forming a sizable fraction of the total membrane proteins, 25%), it is not discussed why these proteins don't resemble their intracellular N-termini counterparts in the translational pause mechanism*. *If the pause is required for prevention of aggregation of the intracellular N-termini proteins, why is it not necessary for the extracellular ones?*

The difference between the two groups of topologies indeed intrigued us. As mentioned, the exclusion of these proteins was done due to small sample size and consequently low statistical power. Although these proteins may show an interesting behavior, at present we feel that we have insufficient data to analyze them separately. It is possible for instance that they have abundant pauses at other locations or that they do not utilize the pause strategy at all, or that they also utilize pauses next to TM2 but coincidently not in the small group of proteins that we analyzed. Without knowing which possibility is correct, we prefer not to speculate and discuss this point further, beyond reporting the intriguing observation. To clarify this issue, we added the sentence: “Therefore, with the current limitations, we cannot conclude if these proteins may have pauses elsewhere or if they do not utilize the pause strategy altogether.”

*3) The following is confusing. The authors first assign the peak of pause to the TM2 of the intracellular N-termini proteins. Yet, when discussing the possible irrelevance of the positive-inside rule in this case it is stated, “the N-termini of these TMs are actually extra-cellular”. One can't figure out whether the subject is still the intracellular N-termini protein or extracellular ones. On a related note, it would be interesting to investigate whether or not any particular type of amino is enriched in the N-termini of the extracellular ones*.

This was indeed confusing. We now completely modified this section and include analysis of amino acid enrichments across the TM for proteins with both extracellular and intracellular N-termini (Figure 2—figure supplement 1). The text was modified accordingly.

Important note:

During the re-analysis for the revision, we noticed a small mistake in Figure 2 (now 2E). In the original manuscript, this figure appears after we note that we will not further analyze proteins with extracellular N-termini; however the original Figure 2 analyzed proteins with both intra- or extra-cellular termini. For the sake of reproducibility we now corrected this such that only proteins with intracellular N-termini are analyzed. This does not qualitatively affect the results, but it increases the fraction of proteins having pause next to TM2 from 22% to 25%.

*4) Where it is stated that “only ∼22% of membrane proteins (69 out of 321) have identifiable pauses at codons -5 to +1 relative to TM2 start”*. *This doesn't seem to be consistent with the earlier statement that 69% of the membrane proteins have at least 1 pause site between codon 16-60. Although it is not clear how many of these membrane proteins have either TM1 or 2 or both, one could infer from*
Figure 1
*that the frequency of pause at TM1 and 2 is more or less 50:50. What could explain the discrepancy between 22% and 34.5% (half of 69%)?*

The discrepancy is explained by the range of codons taken into account in the two cases. In Figure 1, the two peaks in codons 16-60 are divided very roughly to region I and II. Region II spans roughly 21 codons (codons 40-60). In this case, it is indeed expected that at least 35% of the proteins will have pause in region II. Notably, in individual proteins, some of these pauses may not be extremely close to TM2 start. In contrast, in Figure 2 we zero in only on codons very close to TM2 start. Consequently, the 22% percentage (now 25%, see below) relates to a codon range of 7 codons only (codons -5 to +1 relative to TM2 start). Therefore, in this case we may consider a protein as having no pause in this range even if it contains a pause in the third residue of TM2 (i.e., position +2 relative to TM2 start), whereas in the former case (i.e., region II of Figure 1) we may consider the protein as having a pause.

*5) As a negative control in*
Figure 2*, the ribosome densities aligned to TM2 for membrane protein-encoding mRNAs without SD-like sequences should be shown (the ones that were filtered out). Like in*
Figure 6
*for the TM1 aligned sequences*.

This was addressed with the new Figure 2 and new text.